# Calibration-Free Mobile Eye-Tracking Using Corneal Imaging

**DOI:** 10.3390/s24041237

**Published:** 2024-02-15

**Authors:** Moayad Mokatren, Tsvi Kuflik, Ilan Shimshoni

**Affiliations:** The Department of Information Systems, University of Haifa, Haifa 3498838, Israel; moayad.mokatren@gmail.com (M.M.); tsvikak@is.haifa.ac.il (T.K.)

**Keywords:** eye tracking, auto calibration, corneal imaging

## Abstract

In this paper, we present and evaluate a calibration-free mobile eye-traking system. The system’s mobile device consists of three cameras: an IR eye camera, an RGB eye camera, and a front-scene RGB camera. The three cameras build a reliable corneal imaging system that is used to estimate the user’s point of gaze continuously and reliably. The system auto-calibrates the device unobtrusively. Since the user is not required to follow any special instructions to calibrate the system, they can simply put on the eye tracker and start moving around using it. Deep learning algorithms together with 3D geometric computations were used to auto-calibrate the system per user. Once the model is built, a point-to-point transformation from the eye camera to the front camera is computed automatically by matching corneal and scene images, which allows the gaze point in the scene image to be estimated. The system was evaluated by users in real-life scenarios, indoors and outdoors. The average gaze error was 1.6∘ indoors and 1.69∘ outdoors, which is considered very good compared to state-of-the-art approaches.

## 1. Introduction

In daily life, vision is a crucial human faculty. As we move or look around, the ability to see allows us to gather information, some trivial but some essential, about our environment. We use vision to identify objects, recognize individuals, and find our way around. Vision, however, even though three-dimensional, only gives us surface information. If we want more knowledge about what we are looking at, we must stop and seek the information through other media than our eyes. Despite the vast amounts of information available online today, we cannot seamlessly access the data. We must take pictures or submit queries and wait for results. This usually entails interacting with our mobile devices. With all of today’s technological advancements, why should we not be able to do this just by looking at objects of interest?

To enable this technology, we must be able to track an eye’s point of gaze (PoG) as an indicator of human visual attention. Given that eye gaze can indicate the user’s object of interest by identifying the user’s fixation point and translating this into an object in the scene, it is becoming an important input modality. Eye tracking is the process of determining and measuring a user’s line of sight to the object they are observing. It is an active area of research and has many applications in multiple areas including clinical applications [1], surgical training [2], visual systems such as shopping research [3], navigation [4], psychology [5], emotion recognition [6], museum visits [7], and more.

Mobile eye tracking devices have begun moving from the pages of science fiction novels into the marketplace. They have improved in quality, their cost is declining, and several companies such as Pupil Labs [8] and Tobii (www.tobii.com—accessed on 3 January 2024) produce them. Eye gaze is becoming an important input modality since it can convey the object of interest of the user by detecting the user’s fixation point and translating it to an object in the scene. Such devices can be used in varied indoors and outdoors settings. Examples for the potential use of mobile eye tracking as a natural human–computer interaction vary greatly. It can be used to trigger information delivery to users in shopping malls and supermarkets (and in fact in any indoors setting), it can trigger information delivery to tourists in cultural heritage sites and cities (e.g., pedestrians in outdoors urban places), and it can be used in cars safety systems to monitor drivers/pilots awareness, it can be used to guide surveillance drones to areas of interest—to get a better picture of an area observed remotely—whether outdoors or indoors. Knowing the user’s interest, it can easily trigger augmented reality presentation using smart glasses, enhancing what the user is viewing with additional information.

Camera-based eye-tracking systems use infrared (IR) eye cameras and object detection methods to extract eye features (i.e., the center of the pupil) from the captured eye images. The features (points) are transformed afterwards into a front-scene camera’s points that enable the PoG to be estimated. For this to be done accurately, in most tracking systems, users are asked to perform an annoying calibration procedure that makes eye trackers less than user friendly. The most common calibration procedure is to track a screen marker, but other methods also exist [9]. Nevertheless, even if it is calibrated before being used, it is hard to keep a device calibrated in a mobile setting. Whenever a user touches the mobile eye tracker or simply scratches their nose, the device must be re-calibrated because the eye tracker has moved from its initial position.

Numerous studies have employed diverse methods to address the calibration requirement. Alnajar et al. [10] proposed utilizing the gaze patterns of individuals who had previously observed a specific scene as an indication of where a new user might focus when viewing the same scene. They managed to achieve a gaze error exceeding 4∘. Another investigation [11] introduced an online learning algorithm that leveraged mouse-clicked positions to deduce the user’s gaze location on the screen. Each mouse click triggered the acquisition of a learning sample, but the calibration points were obtained using a cursor and its positions. The authors attained an average accuracy of 2.9∘. In a separate study [12], users’ hands and fingertips served as calibration samples, with users simply pointing at various locations in the scene. Although the proposed approach demonstrated accuracy comparable to marker-based calibration techniques, it still necessitated users to direct their gaze to specific points in the scene, achieving an average accuracy of 2.68∘. A more recent investigation [13] utilized an RGBD front camera and saliency-based methods for auto-calibrating a head-mounted binocular eye-tracking system. The reported accuracy was 3.7∘ indoors and 4.0∘ outdoors.

To eliminate the cumbersome calibration procedure, we explored a new method for gaze estimation which is based on corneal imaging [14]. The system we developed uses three cameras, a scene camera and two eye cameras: an IR camera and an RGB camera [15], which enable a 3D perspective. To understand corneal imaging, we must remember that the visible parts of the human eye are the white sclera continuous with the cornea, the iris, and the black pupil, which are all protected by tear fluid. The tear fluid on the cornea turns it into a reflecting surface. To attain a corneal image, the pupil is tracked via the scene camera and then its boundary region is computed. Using the result, the corneal image together with its center (the pupil’s center), we can track the user’s gaze in that image. Nevertheless, since the corneal image is a cropped image of the reflection of the world in the gazer’s eye, it is usually blurred and low resolution. This necessitates the placement of an additional camera, the IR camera, to capture more accurate world scene images (the scenes being reflected but that may be blurry/indistinct). Another challenge is having the RGB eye camera track the pupil continuously, reliably, and accurately. This is a difficult task, especially in harsh lighting conditions and outdoors, where part of the pupil may be occluded as a result of the reflection of the scene.

Deep-learning algorithms undergo training to identify the pupil in both infrared (IR) and RGB images, facilitating the real-time generation of an individualized 3D model of the eye. Following the construction of the 3D model, the algorithm calculates the 3D gaze direction by originating from the center of the eyeball and passing through the pupil center to the external environment. Additionally, this model serves to map the pupil’s position detected in the IR image to its corresponding location in the RGB image and enables the detection of gaze direction in the corneal image. This approach effectively addresses the challenge associated with pupil detection in RGB images, as mentioned earlier.

The present work extends our previous work [15]. Here, we use the device offered in [15], along with the computed 3D model and the 3D-mapping transformation between IR and RGB eye cameras, integrating the per-user point-to-point transformation from the RGB eye camera to the front-scene camera to build a complete calibration-free mobile eye-tracking system. These system components (modeling, 3D-mapping transformation, and integration) and the system as a whole comprise the main contribution of this paper. The system was evaluated in real-life scenarios, indoors and outdoors, and achieved highly accurate and very encouraging results.

## 2. Wearable Eye-Tracking Headset

Our wearable eye-tracking device consists of three cameras assembled on a mobile headset: two eye cameras, one IR camera (800 × 600) and one RGB camera (1280 × 960), and a front RGB scene camera (1280 × 960) (see Figure 1). The IR camera is used to track the pupil continuously and reliably, the RGB eye camera is used to extract and acquire corneal images, and the front RGB scene camera is used to capture the world scene. Detecting and tracking the pupil in the RGB eye camera is challenging, as real-world light reflections can affect the visibility of the pupil in the image drastically (see Figure 2 for an example). Figure 3 presents a snapshot of a user’s eye and the scene in front of the user captured by the three cameras—a bulletin board. The left-most image (Figure 3a) was captured using the IR camera, the center image (Figure 3b) was captured using the RGB eye camera, and the right-most image (Figure 3c) was captured using the front-scene camera. On the one hand, the eye is seen very clearly in the IR image, permitting the deep-learning algorithm to easily detect the pupil with high accuracy. On the other hand, the pupil boundaries are not salient in the RGB image because of the light reflection. This case illustrates the need to use two eye cameras: an IR one for pupil detection and an RGB one for acquiring corneal images.

The mobile headset can be adjusted to fit every person’s height and face shape. The eye cameras (IR and RGB) are placed on a mount with an arm that can be moved forward and backward. The view angle can also be adjusted relative to the user’s eye. The front camera can be adjusted upwards and downwards to fit the world scene view to the user’s height.

Calibrating the device, i.e., the three cameras, is a necessary step to track the eye and to compute the gaze points. The calibration procedure should be carried out before the eye tracker is used, especially when we allow device adjustments that add more degrees of freedom, which is a challenging task that is discussed in Section 4.

## 3. 3D Gaze Estimation

The human eye exhibits two primary axes, namely the visual and optical axes, as depicted in Figure 4. The visual axis is characterized by the line extending from the fovea through the pupil center to the external environment. It is synonymous with the line of sight (LoS) and dictates the true gaze direction. The optical axis, on the other hand, is delineated by the line connecting the eyeball center and the pupil center, extending outward. The angle kappa represents the angular deviation between the visual axis and the optical axis, varying among individuals. During eye tracking, the estimated 3D gaze vector is determined using the optical axis.

Appearance-based methods [16] directly detect and track eye gaze from images, eliminating the necessity for a complete 3D model of the head and eyeball. Instead, these methods develop a mapping function from eye images to gaze directions. They exhibit the ability to handle changes in lighting conditions. Given their typical utilization of the entire eye image as a high-dimensional input feature, they map this feature to a low-dimensional gaze position space. This characteristic makes them potentially effective even with low-resolution images [17]. The pupil center is used as a good approximation indicator for the actual gaze point in the real world (Pupil Core [8], for example). Hence, detecting and tracking the pupil, more specifically, the pupil center, is a crucial step toward estimating the 3D gaze vector.

In our approach, we are interested in tracking the pupil using the RGB eye camera, where the corneal image together with its center (the pupil center) indicate the real gaze point in the world. Then, the pupil center’s image plane, as captured by the RGB eye camera, is transformed into the front-scene camera’s image plane, which generate the gaze points in the scene images.

To estimate the 3D gaze and then build a complete eye-tracking system we need to find and compute the relationship between the three cameras. More specifically, we must transform each point from one camera to the other, moving from the IR eye camera to the RGB eye camera, and from the RGB eye camera to the front-scene camera. Hence, the process of estimating the 3D gaze is divided into several stages:Capturing the pupil using the IR eye camera.Transforming the pupil center’s points from the IR eye camera to the RGB eye camera using a 3D-mapping transformation.Transforming the pupil center’s points from the RGB eye camera to the front-scene camera.

Since the headset must be adjusted for each user, the latter’s face shape must be taken into consideration. The view angle, the distance between the eye cameras, and the user’s eye shape will also vary between users. Hence, each stage of the 3D gaze estimation computation comprises several steps that enable calculation of the relationships according to how the headset is adjusted.

The paper by Mokatren et al. [15] introduces a method for 3D gaze estimation through RGB-IR cameras, eliminating the requirement for calibration. The hardware setup involves a headset equipped with two cameras—an IR camera for pupil detection and an RGB camera for capturing corneal images. The technology is designed to estimate 3D gaze direction without the necessity of individualized calibration procedures. In the auto-calibration process, the software calculates the 3D model of the eye and subsequently determines the 3D gaze direction. Importantly, this process is non-intrusive, and users remain unaware that the system is undergoing self-calibration. In order to create a customized 3D model of the eyeball, it is necessary to gather sample points from both cameras. For this purpose, both cameras capture images of the eye simultaneously. Points are then gathered from the detected pupil boundaries using a deep-learning algorithm. With a set of corresponding points in the two image planes, examining the intersections of the projected rays from the corresponding pair yields a collection of 3D points. These points, known to lie on the surface of the eyeball, allow for the determination of the eyeball’s center, given the known radius of 12 mm, provided a sufficient number of 3D points are available. This study builds upon prior research, utilizing the 3D gaze estimation framework to compute the 3D transformation between eye camera images without the need for calibration. The intricate details of the transformation between RGB eye camera and front-scene camera images are expounded upon in Section 4.3.

Our approach employs a deep-learning algorithm in conjunction with the IR camera to detect the center of the pupil. Subsequently, geometric computation is applied to derive the 3D-mapping transformation. The primary purpose of this transformation is to convert a point in the IR eye camera image plane into a corresponding point in the RGB eye camera image plane. The RGB eye camera serves as the key function of capturing corneal images around the specified point, facilitating their alignment with points in the front scene image and identification of the user’s point of interest.

In contrast to conventional setups utilizing only an eye camera and a scene camera, our method incorporates three cameras. This includes two eye cameras, enabling stereo vision for modeling the eyeball. The computation of the 3D gaze direction initiates from the center of the user’s eyeball and extends through it to the external environment (refer to Figure 5 for an illustration of the concept). Stereo vision is instrumental in modeling the eyeball, and subsequently, the 3D gaze direction is calculated using the eye cameras. The outcome is then transformed into a point within the scene camera’s image.

It is important to note that our system does not necessitate a 3D gaze direction but focuses solely on obtaining a gaze point in the front-scene camera. As a result, we utilize the 3D transformation exclusively for mapping points between eye cameras. In Section 4.3, we elaborate on how we accomplish this transformation between planes, specifically from the eye camera to the front camera, using image matching.

Points sampled from the detected pupil center in both types of images are employed to construct a 3D model of the user’s eyeball. Subsequently, this model plays a crucial role in calculating the 3D transformation between the two eye cameras and determining the 3D gaze direction. Consequently, when a pupil is detected in an IR image, its corresponding position in the RGB image is computed. Following this, the corneal image is extracted around the detected pupil, essentially delineating its bounding box.

## 4. Calibration-Free Mobile Eye Tracking

In this section, we detail our innovative calibration-free mobile eye-tracking technology. The primary objective is to continuously, reliably, and, most importantly, unobtrusively track the user’s eye and compute gaze points. This involves the user wearing the headset while the device autonomously calibrates, eliminating the need for users to focus on target points or engage in specific tasks. Our primary focus is on calibrating the headset’s front camera, enabling the inconspicuous computation of a transformation between the RGB eye camera and the front-scene camera.

Illustrated in Figure 6, our calibration-free mobile eye-tracking system initiates by employing a deep-learning algorithm to track the pupil in the IR eye camera. The algorithm computes the pupil center along with the pupil’s bounding box, representing the corneal image. The points derived from the pupil center and its bounding box undergo transformation to the RGB eye camera image plane through the 3D-mapping transformation. Subsequently, the 3D gaze vector is computed based on the user’s unique eyeball model. This results in having, at every moment, a corneal image with a corresponding gaze point and a 3D gaze vector extending from the eyeball center through the pupil center to the external environment. The 3D gaze vector translates to an actual gaze point in the world, representing the user’s point of interest.

Ultimately, the point of interest in the world undergoes transformation into the front scene image via a matching points-based transformation (refer to Section 4.3 for details). This comprehensive process ensures continuous and unobtrusive eye tracking with the ability to compute gaze points in real time.

### 4.1. One-Time Offline Calibration

Our stereo system consists of both IR and RGB cameras. The goal is to seamlessly convert a point located in the IR camera image plane to its corresponding position in the RGB camera image plane. Typically, conventional 2D transformations, like the fundamental matrix, transform a point into a line. However, in our specific scenario, we employ the eyeball model, which enables us to transform a point directly into another point. The proposed 3D-mapping transformation, as detailed in [15], precisely accomplishes this task. Our approach relies solely on the 3D-mapping transformation between the IR and RGB cameras. Gaze estimation, outlined in Section 3, follows a distinct methodology. For the computation of point-to-point transformation, essential information is required, including the internal parameters of the cameras and details about the focused object (the eyeball). The average human eyeball radius, drawn from the existing literature, is approximately 12 mm [18].

Moreover, to compute a 3D transformation between the cameras in a stereo system, knowledge of the relative pose between the eye cameras and the relative pose between the eye cameras and the human eye is essential. The relative pose between the cameras is determined through stereo camera calibration, which is a one-time offline process and is independent of user-specific features. In our system, calibration was executed using a chessboard pattern with 2 mm sized cells, following the technique proposed in [19].

For the evaluation purposes discussed in Section 5.2, the front-scene camera underwent an additional offline calibration procedure—in addition to using the chessboard. The camera’s internal parameters (calibration output) are used mainly for measuring the distance between target points in the real world. We used the front-scene camera parameters to compute, compare, and measure differences between computed gaze points and ground truth points.

### 4.2. Corneal Images Acquisition

Our main goal is to track the user’s gaze. To do this, following the appearance-based method for gaze estimation, we track the pupil and compute the pupil’s center. We are interested in detecting the pupil in the RGB eye camera where the scene of the world is reflected onto the cornea. Since this is challenging, as the detection is affected by lighting conditions, especially outdoors and in harsh lighting conditions, we detect the pupil using the IR eye camera. The camera provides a reliable image of the pupil whose bounding box is then transformed to the RGB eye camera’s image plane using the 3D-mapping transformation [15] (see Figure 7, left). We can extract the corneal image by cropping the RGB eye image around the pupil using the transformed bounding box. Since the eye acts as a mirror, we flip the corneal image to match the world scene (see Figure 7 center—the red dot is the gaze point).

Because image resolution is very limited in corneal imaging systems, we should track and compute the user’s gaze via the front camera. For this, we need to transform the gaze point (from the RGB eye camera) to the front-scene camera to detect gaze points over high-resolution images of the world scene, as discussed earlier (Section 3) and illustrated in Figure 6.

### 4.3. Gaze Transformation to the Front-Scene Camera

To build a complete mobile eye-tracking system, we need to compute gaze points in the front-scene camera image plane. In this section, we present our 2D point-to-point transformation method. The aim is to transform a point from an image taken by the RGB eye camera to a point in an image taken by the front-scene camera.

Since the RGB eye camera and the front-scene camera are placed opposite each other, they do not face the same scene. The RGB eye camera faces the eye and the front-scene camera faces the world. Hence, our challenge is to compute a transformation (relationship) between two cameras that are facing different views. Several works tried to calibrate omnidirectional camera systems by using different approaches such as calibrating using mirrors [20,21]. Still, these approaches require special setups and are limited in the way the cameras should be fixed in their relative positions. Our headset allows camera adjustments, so current methods for calibrating the cameras do not align with our needs.

In our approach, we exploit the fact that the world is reflected onto the eye’s surface (cornea). Looking at the corneal images, we can see the world scene (as seen by the front camera also) reflected in the eye (see Figure 7 for an example). Hence, at every moment we have two images of the world scene; one is captured normally by the front-scene camera and another (corneal image) acquired by the RGB eye camera, as described in Section 3.

Our goal is to consistently transform the gaze point from the RGB eye camera into a corresponding point in the front-scene camera image in real time. In essence, we aim to calculate a point-to-point transformation between the corneal image and the scene image captured by the front-scene camera. To achieve this, we employ a 2D transformation by aligning and matching the two images. During the image-matching process, we leverage local image features that remain robust in the presence of nearby clutter or partial occlusion. These features exhibit at least partial invariance to factors such as changes in illumination, 3D projective transforms, and common object variations. The utilization of such features ensures a reliable and accurate point-to-point transformation between the corneal and scene images.

Various types of local image features have been developed, with SIFT (Scale-Invariant Feature Transform) being the most popular [22]. Other types include SURF (Sped-Up Robust Features) [23] and ORB (Oriented FAST and Rotated BRIEF) [24]. SIFT features are known for their invariance to image scaling, translation, and rotation, as well as partial invariance to illumination changes and affine or 3D projection.

When the SIFT algorithm is applied to an image, it generates a set of features along with their descriptors. Matching these descriptors between images allows the computation of a set of potential matches between corresponding features. However, in cases like ours, where we are dealing with a low-resolution image (the corneal image) containing a limited number of descriptors, matched scenes may still be prone to errors.

To address this challenge, we leverage the geometric relationships between the positions of matched features in the two images using a homography matrix. The homography matrix is computed using a robust estimation procedure from the RANSAC (Random Sample Consensus) family [25]. Through the application of a homography matrix, we can effectively transform points from one image plane to another, providing a solution to the matching problem and contributing to the accuracy of our point-to-point transformation.

In this work, we compute the point-to-point 2D transformation between the corneal images and the front-scene camera by matching SIFT [22] features computed from both corneal images and the front-scene camera and computing the homography matrix. Figure 8 and Figure 9 show matched SIFT features in two different scenes between the RGB eye image and front scene image. In our method, to compute the homography matrix, we take into consideration only features placed in the corneal image (around the pupil). We repeat the SIFT feature extraction procedure using a lot of pairs of images and take into consideration only the correct ones using RANSAC.

The gaze point in the world scene is the matched point of the pupil’s center in the corneal image. Hence, to compute the gaze point, we want to transform the pupil center to the front camera using the homography matrix that matches points to points in the two images planes. Let *H* to be the homography matrix, and let (*u*,*v*) be the pupil center coordinates in the corneal image. The relationship between the point and its corresponding point ($u^{\prime }$,$v^{\prime }$) up to factor *S* is
$$S*\left(\begin{array}{c} u^{\prime }\\ v^{\prime }\\ 1 \end{array}\right)=H*\left(\begin{array}{c} u\\ v\\ 1 \end{array}\right).$$

The matched point of the pupil center in the front camera ($u^{\prime }$,$v^{\prime }$) is computed as follows:$$\left(\begin{array}{c} u^{\prime }\\ v^{\prime }\\ 1 \end{array}\right)=\left(H\times \left(\begin{array}{c} u\\ v\\ 1 \end{array}\right)\right)/S.$$

## 5. Experimentation

The experiment was intended to evaluate our calibration-free mobile eye-tracking technology in different real-world scenarios, indoors and outdoors. We were interested in evaluating the efficiency of the auto-calibration procedure and the accuracy of the gaze estimation.

We reiterate that our proposed system does not require any action on the part of the user for device calibration. The user simply puts on the device and behaves and gazes normally. For the experimentation, we asked the participants to perform normal, everyday activities (such as looking at objects or screen markers or watching a video).

### 5.1. Tools and Methods

We have successfully developed a prototype that autonomously calibrates the headset based on the user’s individual characteristics, specifically adjusting to the user’s head. As depicted in Figure 1, the user wears the device, and the prototype systematically constructs a 3D model of each user’s eye. Subsequently, it computes robust and reliable 3D transformations between the eye cameras. Finally, the gaze point is seamlessly transformed to the front-scene camera.

The entire process of building the 3D eye model is carried out automatically, requiring no active participation from the user other than wearing the device. This ensures a user-friendly and unobtrusive calibration procedure, where the technology adapts to each user’s unique characteristics effortlessly.

The development of the prototype involved utilizing both MATLAB and Python running on a laptop (Core i7-5600U-16 GB). The training procedure for the deep learning pupil detection, as described in [15], was conducted using Python, specifically leveraging the TensorFlow API for object detection [26]. The process of capturing images from the cameras and implementing the overall framework was executed in MATLAB, which also involved the integration of the trained deep learning pupil detection model.

The cameras’ capture tool was designed to run two parallel processes, capturing images simultaneously from both the infrared (IR) and RGB cameras, maintaining a frame rate of 20 frames per second (fps). To avoid an excess of duplicate eye images within a short timeframe, a capture rate of 1 fps was employed. Notably, to ensure synchronization, a pair of eye images (from the IR and RGB cameras) with the nearest timestamps was selected every second. This synchronization approach enhances the accuracy and reliability of the captured eye image pairs. The whole prototype system achieves an 8 fps computation time.

### 5.2. Participants and Experimental Procedure

The experiments were conducted in a realistic setting in two different scenarios, indoors and outdoors. The experiment had 12 participants (5 females, 7 males). Each participant took part in 2 sessions, which meant that the experiment comprised 24 sessions. The experiment was intended to evaluate the feasibility and accuracy of our system under different conditions. The experiment took about 30 min and was divided into 3 sections: (1) data collection indoors, which was used for auto-calibrating the system; (2) tracking screen markers shown on a PC screen indoors; and (3) looking at objects outdoors. Participants, monitored by a researcher, wore the mobile headset connected via a wire to a laptop and used the system. They were instructed to behave normally and gaze naturally during each session.

#### 5.2.1. Data Collection for Auto-Calibration

The aim of this part of the indoor session was to collect data from the three cameras to auto-calibrate the system (Task 1). The data collection enabled us to compute the 3D-mapping transformation between the eye cameras and to compute the point-to-point transformation from the RGB eye camera to the front-scene camera. During this part of the session, participants performed two tasks: (1) they walked freely in an indoor environment and gazed naturally at objects and (2) sat normally and watched a 2-min video presented on a 24” PC screen. Corneal images were captured by the cameras’ capture process.

A total of 100 triads of images from the 3 cameras were collected during the first task (walking freely indoors). Pairs from the IR eye camera and the RGB eye camera were utilized to build the 3D eyeball model used for computing the 3D-mapping transformation between the IR and RGB eye cameras, as described in [15]. Figure 3 shows a sample of collected images when a participant walked freely in an indoor environment while looking at a bulletin board.

A total of 120 triads of images from the 3 cameras were collected during the second task (watching a video). A total of 100 pairs from the RGB eye camera and from the front-scene camera, together with the 100 pairs (220 in total) of RGB eye images and the front-scene camera that were collected in the first task (walking freely indoors), were used to compute the transformation (homography) between the RGB eye camera and the front-scene camera. From each pair (see Figure 10 for an example), SIFT descriptors were computed and stored. All SIFT descriptors from the 220 pairs of images were used to compute the mapping transformation between the RGB eye camera and the front-scene camera.

#### 5.2.2. Eye-Tracking Screen Markers

The aim of this section of the indoor session was to collect data from the three cameras while participants tracked screen markers on a computer screen (see Figure 11 and Figure 12). This method is widely used to evaluate eye-tracking techniques [10,11,12,13]. The participants were asked to sit normally in front of the screen and track the screen marker. The screen markers changed their position in the grid randomly. Each marker remained in a grid cell for three seconds while flashing to attract the participant’s attention and make them look at it. A total of 80 triads of images from the 3 cameras were used for evaluation purposes. From each triad, we computed the gaze point (point on the front scene image plane), which was used later to compute gaze errors.

#### 5.2.3. Eye Tracking Outdoors

The aim of this session, a test of the system’s feasibility, was to collect data from the three cameras while participants who were outside looked at objects. This was considered a challenging undertaking. Participants were asked to look at both nearby objects (1 to 3 m), such as a sign, and far objects that the accompanying researcher pointed at (20 m and more), such as a house or university campus building. See Figure 13 and Figure 14 for examples. For far objects, the researcher stated loudly which object he/she was indicating, mainly to ensure that the participant looked at the object. A total of 100 triads of images from the 3 cameras were used for evaluation purposes. From each triad, we computed the gaze point (point on the front scene image plane), which was used later to compute gaze errors.

### 5.3. Evaluation

The main parts of the evaluation assess the accuracy of the proposed calibration-free mobile eye-tracking technology under different conditions. As noted in Section 5.2.2 and Section 5.2.3, each participant took part in two experimental sessions: (1) indoors during auto-calibration and tracking of markers on screen, and (2) looking at objects outdoors.

Triads of images of the eye and the world view were collected during the two sessions. A total of 180 triads—80 triads of images while tracking screen markers and 100 triads of images while looking at objects outdoors—were documented. For each triad, we computed the gaze point in the front scene image automatically using our system prototype and using the mapping transformation to the front camera described in Section 4.3.

To compare the gaze accuracy, we tagged the point of interest (screen marker center and outdoor object center) in each front scene image in every test triad manually, which is the best way possible to reach the ground truth. We used the gaze angle error between the computed gaze point and the manually acquired gaze point as a metric for comparison. The angle error between the two gaze points was computed using the front-scene camera’s internal parameters (described earlier in Section 4.1) and multi-view geometry.

### 5.4. Results

A total of 2160 triads of images were used in the evaluation (from 12 participants, each with 180 triads from two sessions). For each triad of test images (IR eye image, RGB eye image, and front scene image), we declared triad a valid test triad if the gaze was manually tagged. The resulting metric is the angle error in degrees. For each participant, we calculated the median of gaze errors to filter out outlier measurements. Table 1 summarizes the results of the indoor session and Table 2 summarizes the results of the outdoor session. On average, the gaze error in degrees was 1.67∘ indoors and 1.69∘ outdoors.

## 6. Discussion

We propose a novel calibration-free mobile eye-tracking approach. Our approach was implemented and evaluated in two different realistic scenarios, indoors and outdoors. The system auto-calibrates the device unobtrusively and without involving the user in the process.

The system achieved an average gaze error of 1.67∘ (SD = 0.87) indoors and 1.69∘ (SD = 0.7) outdoors. The accuracy is considered very good compared to the state-of-the-art methods that try to compute gaze without needing calibration under similar conditions and with a similar number of participants. Moreover, our system uses a more accurate ground truth comparison (manually labeled points) method than other systems. Table 3 presents a comparative analysis with state-of-the-art methods. Our system’s versatility was showcased through successful operation in various real-world environments, both indoors and outdoors. Additionally, our automatic and non-obtrusive calibration technique eliminates the need for users to undertake specific actions. This is in contrast to existing methods that necessitate user participation in the calibration process, including the use of screen or board markers.

In the development of the proposed system, we used the existing framework for computing the 3D eye model and the 3D-mapping transformation between the IR eye camera and the RGB eye camera [15]. The framework suffers from technical limitations, mainly the effect of harsh light when attempting to detect the pupil in RGB eye cameras for computing the model and dealing with special cases such as dark irises. These challenges may still occur in our system but can be dealt with as discussed in [15].

The main limitation of the proposed system is the need for the data collection used for building the eye model and computing the mapping transformation to the front camera. In our approach, we do this unobtrusively without asking the user to look or track objects. In situations where the mobile eye tracker is used on a daily basis, individually, the need for modeling the user’s eye may be eliminated, as the model will be created once and used continuously.

The main limitation of the user study was finding a world scene where we could detect a large number of features in the corneal images used for computing the mapping transformation to the front camera. For that, we presented a cartoon video on a screen where a lot of image features were reflected onto the eye. In a real system, the same approach for computing the mapping transformation can be used, but it may take a bit longer until sufficient features have been collected.

The experiment included only 12 participants, which is considered a limitation. On the one hand, it was hard to recruit participants and collect data in the wild. On the other hand, the scope of our experiment is in line with previous works in the literature (please see Table 3). We will consider expanding the experiment in the future.

The experimental scenario, in realistic settings (not in the lab), is considered a limitation as well. We chose a scenario where we could easily recruit participants. Given the limitations, we asked the participants to look at different objects from different distances (a flower, a window, a building, and more).

The advancement made in mobile eye-tracking technology may help many studies that use eye gaze as an indicator of human visual attention. Still, current technology suffers from two main limitations: the need for calibration and the unavailability of a reliable system for outdoor eye gaze tracking. In this work, we provide a calibration-free mobile device that detects the human gaze and can be used in different scenarios, indoors and outdoors. The device can also be used in different disciplines: interaction with real or virtual environments, the metaverse, cockpit control, and more.

## 7. Conclusions

In this work, we presented and evaluated a calibration-free mobile eye-tracking system. The system uses a mobile device consisting of three cameras, an IR eye camera, an RGB eye camera, and a front-scene camera. The system auto-calibrates the device unobtrusively and without involving the user in the process. The user is not required to follow special instructions to calibrate the system. The IR eye camera is used to track the pupil continuously and reliably and to compute the gaze point. The RGB eye camera is used to acquire corneal images and the front-scene camera is used to capture high-resolution scene images. A per-user 3D model of the eye is built to compute the 3D transformation between the IR and RGB eye cameras, and a point-to-point transformation between the RGB eye camera and the front-scene camera, unique to the user, is computed.

The proposed system does not require any initial calibration procedure. The user can simply put the eye tracker on and start moving around and using it, as the calibration procedure is automatic and unobtrusive. The system was evaluated in an experiment in real-environment scenarios, indoors and outdoors, and the results are promising. The system achieves very low gaze errors: on average 1.67∘ indoors and 1.69∘ outdoors. The system can be used in a variety of mobile scenarios, indoors and outdoors with high accuracy. Future work will focus on integrating the proposed calibration-free mobile eye-tracking system into information delivery systems, where gaze is used as an intuitive pointing device.

## Figures and Tables

**Figure 1 sensors-24-01237-f001:**
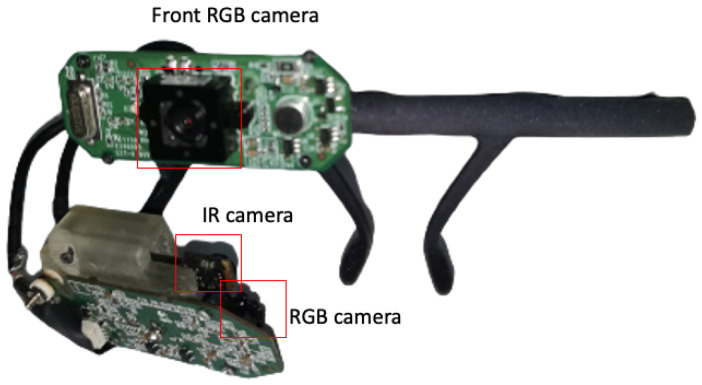
Front view of the eye-tracker headset with key components.

**Figure 2 sensors-24-01237-f002:**
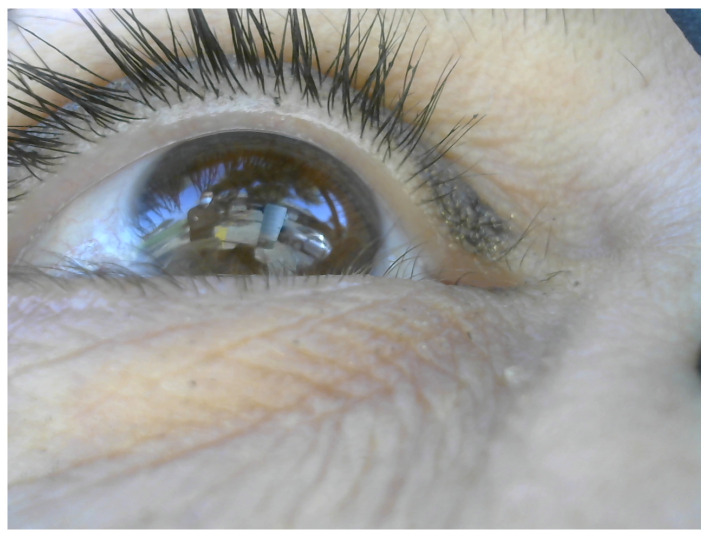
RGB eye camera snapshot with light reflection. The boundary of the pupil is invisible.

**Figure 3 sensors-24-01237-f003:**
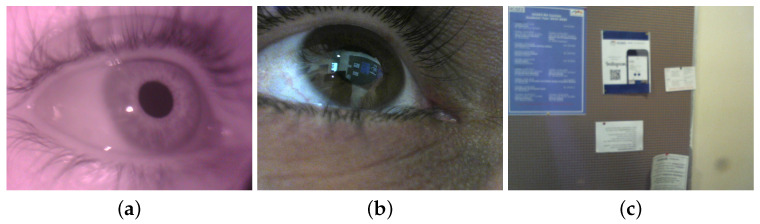
(**a**) IR camera image of the eye. (**b**) RGB camera image of the eye. (**c**) Image as seen by user.

**Figure 4 sensors-24-01237-f004:**
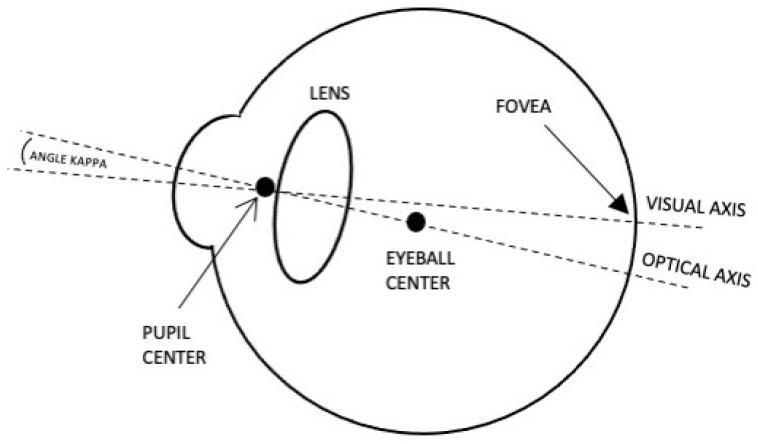
Human eye visual axes. Both visual and optical axes pass through the pupil center. Only the optical axis can be computed.

**Figure 5 sensors-24-01237-f005:**
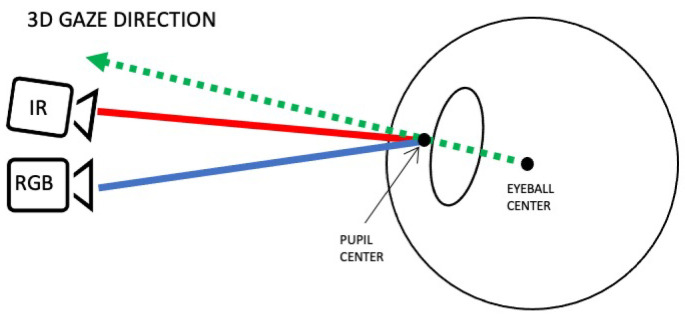
Illustration of 3D gaze direction mapping using a 3D model of the eye. The green ray is the 3D gaze direction.

**Figure 6 sensors-24-01237-f006:**
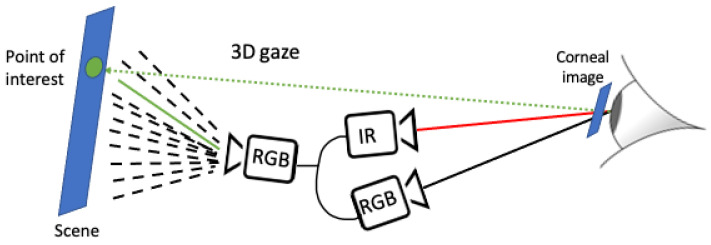
Illustration of gaze computation. The green dashed ray is the 3D gaze direction. The green point is the gaze point in the world. The green ray is the reflected vector of the gaze point into the front-scene camera.

**Figure 7 sensors-24-01237-f007:**
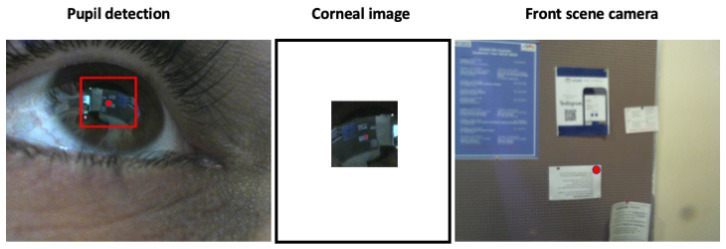
Corneal images acquisition process. The left-most image illustrates the pupil detection in the RGB eye camera (bounding box and gaze point). The center image is the acquired corneal image (flipped) with a red gaze point. The right-most image is the front-scene camera image with the gaze point (point of interest).

**Figure 8 sensors-24-01237-f008:**
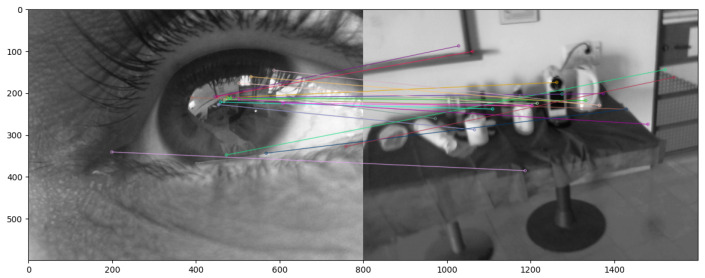
Matching the SIFT features of the RGB eye image and the front scene image while looking at a table with a coffee machine.

**Figure 9 sensors-24-01237-f009:**
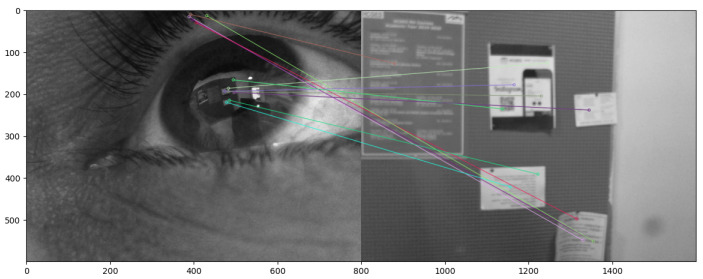
Matching the SIFT features of the RGB eye image and the front-scene image while looking at a bulletin board.

**Figure 10 sensors-24-01237-f010:**
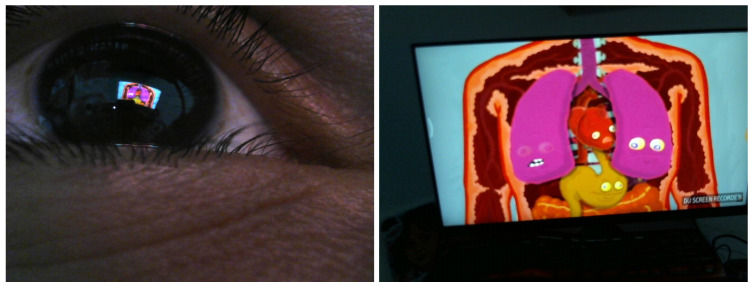
Snapshot from RGB eye camera (**left**) and front-scene camera (**right**) while watching a video on a PC screen.

**Figure 11 sensors-24-01237-f011:**
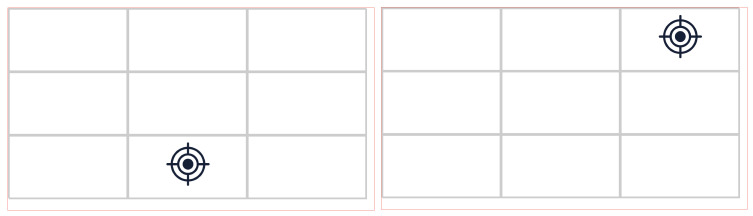
Two screenshots of the screen markers in different positions shown on a PC screen.

**Figure 12 sensors-24-01237-f012:**
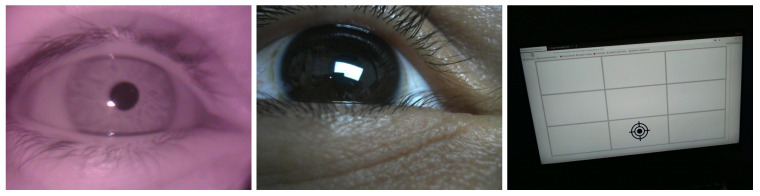
Sample of triad of images captured by the three cameras while tracking a screen marker.

**Figure 13 sensors-24-01237-f013:**
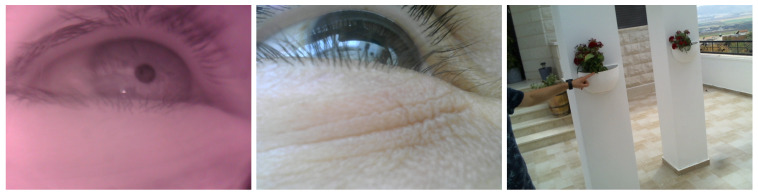
Snapshot of triad of images while the participant looked at a nearby object (a flower). We can also see the researcher pointing at the desired object of interest.

**Figure 14 sensors-24-01237-f014:**
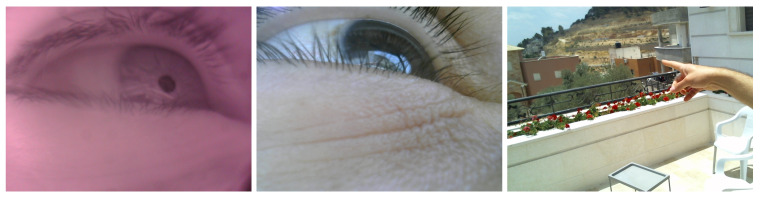
Snapshot of triad of images while the participant looked at a far away object (a house). We can also see the researcher pointing at the desired object of interest.

**Table 1 sensors-24-01237-t001:** Summary of the gaze errors in degrees for indoors session per participant.

	# 1	# 2	# 3	# 4	# 5	# 6	# 7	# 8	# 9	# 10	# 11	# 12
Mean	2.01	2.18	1.63	1.68	1.02	1.75	1.66	1.58	1.50	1.59	1.71	1.68
STD	1.21	0.97	0.80	1.03	0.71	0.73	0.80	1.10	0.65	0.70	0.71	1.05
Median	1.80	2.03	1.49	1.33	0.78	1.54	1.47	1.19	1.24	1.60	1.53	1.34

**Table 2 sensors-24-01237-t002:** Summary of the gaze errors in degrees for outdoors session per participant.

	# 1	# 2	# 3	# 4	# 5	# 6	# 7	# 8	# 9	# 10	# 11	# 12
Mean	2.27	1.42	1.95	2.14	1.08	1.78	1.84	2.11	1.93	1.39	1.71	1.96
STD	0.69	0.74	0.76	0.57	0.65	0.64	0.92	0.60	0.75	0.70	0.61	0.75
Median	2.31	1.24	1.84	2.17	1.00	1.66	1.59	2.00	1.71	1.32	1.65	1.73

**Table 3 sensors-24-01237-t003:** Summary of comparison with state-of-the-art methods.

Work	Our Work	[12]	[11]	[13]	[10]
Average error (deg)	1.67–1.79	2.68	2.9	3.7–4.0	4.2
SD	0.87–0.7	0.67	2.1	N/A	1.3
#Participants	12	12	10	10	10
Target	Screen markers and outdoor objects	Screen markers	Screen markers	Board with markers	Screen markers
Ground truth	Manually hand labeled	Pupil eye tracker	Tobii TX300	Similar RGB-D methods	Tobii T60XL

## Data Availability

The data in this experiment is unavailable due to privacy restrictions.

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
