# Peer review of "Calibration-Free Mobile Eye-Tracking Using Corneal Imaging"

_sensors, 2024, doi:10.3390/s24041237_

Round 1

Reviewer 1 Report

Comments and Suggestions for Authors

In this paper, authors have presented and evaluated a calibration-free mobile eye-tracking system. The system uses a mobile device consisting of three cameras, an IR eye camera, an RGB eye camera, and a front scene camera. The system auto-calibrates the device unobtrusively and without involving the user in the process. The IR eye camera is used to track the pupil continuously and reliably and to compute the gaze point. The RGB eye camera is used to acquire corneal images and the front scene camera is used to capture high-resolution scene images. 

The paper is written very well and is technically very sound. However, authors have not mentioned the motivation for developing this system. What real-life scenarios/ practical applications motivated the authors to develop this system and algorithm?

Reviewer 2 Report

Comments and Suggestions for Authors

The authors should provide more technical results based on the given theoretical results. Also, this manuscript lacks statistical analysis of the proposed method and comparison results before reviewing this manuscript. Otherwise, it is good. 

Comments on the Quality of English Language

Needs a minor edition. 

Reviewer 3 Report

Comments and Suggestions for Authors

The authors present an interesting eye-tracking method that includes auto-calibration.

While the method is interesting, the paper can be improved upon. My feeling is that the authors reiterate the same statements many times throughout the paper, instead of going into further details on other topics. This is due to poor sectioning of the explanation and not adhering to the section topic.

For example, the issues of pupil detection from RGB and benefits of using IR camera are discussed in sec. 2 and then again in sec. 3, second paragraph. Then, the next paragraph starts with "In our approach, the IR eye camera is used to track the pupil..." - essentially reiterating the same information.

I recommend restructuring of the text to clear sections where only one part of the proposed method is discussed in-depth.

Additionally, I was missing some key details in the explanation:

1) Hardware description - should also include camera resolutions, which is missing from the text completely.

2) Eye-model building - should include a better description of the model parameters and how they are estimated from the images

Minor issues:

Fig. 1 should include some description markers denoting key components of the device.

Fig. 8 & 9 seem to have swapped descriptions (board vs table).

Lastly, I would like an addition of computational complexity of the system. What kind of computational hardware is required and what were runtimes of individual steps on such hardware?

Reviewer 4 Report

Comments and Suggestions for Authors

In the article, a method for tracking the point of view which does not require calibration is presented. The method is based on the use of three cameras (two RGB and one IR). One camera monitors the position of the center of the pupil, another monitors the image of the scene on the cornea, and the third monitors the scene. It is claimed that based on the ratio of the three images, the system can self-calibrate and track the point of view. Such a statement can be accepted, although it remains unclear how the auto-calibration procedure takes into account the geometric features of the individual's eye and face. The statement “The pupil center is used as a good approximation indicator for the actual gaze point in the real world” (142) in my opinion requires clarification. Perhaps the error that occurs when determining the point of view at the center of the pupil is not large, but it would be good to confirm this. Below it is stated that “The gaze point, which is the center of the corneal image, is the point of interest in the real world reflected on the cornea (mirrored image) and matches a point of interest in the real world” (264). This does not correspond to the statement above. This should be clarified. It would be good to provide a more detailed justification for the applicability of the presented approach. Perhaps a diagram showing the path of the rays at various turns of the eye and head would clarify the situation.

There is text repetition from 98 to 110 and from 111 to 123

Round 2

Reviewer 2 Report

Comments and Suggestions for Authors

The revised version of this manuscript is satisfactory after inserting the following reference related to image analysis.

Nonlinear Dynamics 74, (2013) 1169–1181

Comments on the Quality of English Language

needs small English corrections

Author Response

We studied the suggested paper that the reviewer asked us to cite and  could not see how this paper is related to our paper.

As far as we can tell our paper does not describe a chaotic system.

Could the reviewer please explain where and why to cite this paper.